# Anomalous polarization enhancement in a van der Waals ferroelectric material under pressure

Xiaodong Yao[1,4], Yinxin Bai[1,4], Cheng Jin[2,4], Xinyu Zhang[1], Qunfei Zheng[1], Zedong Xu[1], Lang Chen [1], Shanmin Wang [1], Ying Liu[1], Junling Wang [1,3] ✉ & Jinlong Zhu [1] ✉

$CuInP_2S_6$ with robust room-temperature ferroelectricity has recently attracted much attention due to the spatial instability of its Cu cations and the van der Waals (vdW) layered structure. Herein, we report a significant enhancement of its remanent polarization by more than 50% from 4.06 to 6.36 $\mu C\ cm^{-2}$ under a small pressure between 0.26 to 1.40 GPa. Comprehensive analysis suggests that even though the hydrostatic pressure suppresses the crystal distortion, it initially forces Cu cations to largely occupy the interlayer sites, causing the spontaneous polarization to increase. Under intermediate pressure, the condensation of Cu cations to the ground state and the polarization increase due cell volume reduction compensate each other, resulting in a constant polarization. Under high pressure, the migration of Cu cations to the center of the S octahedron dominates the polarization decrease. These findings improve our understanding of this fascinating vdW ferroelectric material, and suggest new ways to improve its properties.

Ferroelectricity is characterized by a spontaneous polarization below the Curie temperature, which can be switched by external electric fields. While the Landau–Ginzburg–Devonshire (LGD) phenomenological theory is simple and useful to describe ferroelectric phase transitions[1,2], modern theory based on the Berry phase allows for accurate calculation of the spontaneous polarization[3,4]. Previously, perovskite oxide ferroelectrics, such as $PbTiO_3$ and $BaTiO_3$, were mostly studied and widely used. To facilitate integration with modern electronics, thin films (both on a substrate and free-standing) were extensively investigated. However, structural and chemical incompatibility at the interface/surface often hamper the ferroelectric properties and device performances. Recently, the development of two-dimensional (2D) van der Waals (vdW) ferroelectric materials opens a new paradigm in the field. 2D vdW ferroelectrics, such as SnTe, $In_2Se_3$, $CuInP_2S_6$ (CIPS), and transition metal chalcogenides ($WTe_2$ and $MoTe_2$), have been reported. Chang et al.[5] discovered a stable in-plane

spontaneous polarization in atomic-thick SnTe down to a 1-unit cell (UC) limit in 2016. Subsequent ab initio calculations suggested a polarization of $2.3 \times 10^{-6}\ \mu C\ cm^{-1}$ for both 2- and 6-atomic-layer γ-SnTe films[6]. Room-temperature ferroelectricity was discovered in CIPS with a transition temperature of ~320 K (Liu et al.[7]), and switchable out-of-plane polarization was observed in thin CIPS of ~4 nm[7,8]. In 2018, Xue et al. reported that room-temperature ferroelectricity exists in hexagonal layered α-$In_2Se_3$ nanoflakes down to the monolayer limit[9]. The calculated out-of-plane and in-plane spontaneous polarizations were 0.97 and 8.0 $\mu C\ cm^{-2}$, respectively[10]. Furthermore, two- or three-layer $WTe_2$ was shown to exhibit spontaneous out-of-plane polarization of ~0.03 $\mu C\ cm^{-2}$[11]. It was also reported that AB-stacked bilayer BN possesses a spontaneous polarization of 0.68 $\mu C\ cm^{-2}$[12].

In general, the out-of-plane spontaneous polarization of 2D vdW ferroelectrics is relatively small, which may hinder their applications in electronic devices. Thus, polarization enhancement in vdW

[1]Department of Physics, Southern University of Science and Technology, Shenzhen 518055, China. [2]Center for High Pressure Science and Technology Advanced Research (HPSTAR), 100094 Beijing, China. [3]Guangdong Provincial Key Laboratory of Functional Oxide Materials and Devices, Southern University of Science and Technology, Shenzhen, China. [4]These authors contributed equally: Xiaodong Yao, Yinxin Bai, Cheng Jin. ✉e-mail: jwang@sustech.edu.cn; zhujl@sustech.edu.cn

ferroelectrics is of crucial importance for both fundamental study and technological development of new ferroelectricity-based electronic devices. Herein, we report a hydrostatic-pressure-driven 56.5% enhancement of the spontaneous polarization of CIPS at room temperature, which is opposite to the usual pressure induced suppression of ferroelectricity[13–16]. For example, the spontaneous polarizations of PbTiO$_3$ and BaTiO$_3$ at room temperature were totally suppressed under pressures of 10 GPa and 2 GPa, respectively[17,18]. Though it has been reported that the ferroelectricity of multiferroic CuCrO$_2$ and TbMnO$_3$[19,20] enhances within a certain pressure range, they were attributed to pressure-induced magnetoelectric phase transitions. The enhancement of remanent polarization in vdW ferroelectric CIPS is another example worthy of further investigation. Detailed Raman analysis suggests that the anomalous behavior is due to an increase in the interlayer coupling upon reducing the vdW gaps, promoting the Cu occupancy at the interlayer sites. Recently, Ming et al.[21] reported an alternative switching method for CIPS using flexoelectric effect. In their experiments, strain gradient drives the migration of Cu cations. However, in this study, hydrostatic pressure is applied, promoting more Cu cations to occupy interlayer sites, resulting in the overall enhancement of polarization.

## Results

### Polarization-electric field (P-E) hysteresis loops of CIPS under pressure

The crystal structure of CIPS is defined by the sulfur framework as shown in Fig. 1a, where the octahedral voids are filled with the triangularly arranged Cu, In, and P-P cations. Bulk crystals are composed of vertically stacked layers weakly linked by vdW interactions. Because of the sites exchange between Cu and P-P pairs from one layer to another, a complete unit cell consists of two adjacent layers, which is required to describe the material's symmetry[22]. When the temperature drops below $T_c$, a first-order phase transition occurs, and the symmetry reduces from C$_{2/c}$ to C$_c$[23]. The ferroelectricity of CIPS originates from the spatial instability of Cu cations. Monovalent Cu cations favor lower coordination because of the second-order Jahn-Teller coupling

between the filled 3$d^{10}$ manifold and the empty 4$s$ orbital. The electric dipoles produced by Cu cations deviating from the center of the S octahedron leads to the macroscopic polarization.

You et al.[24] and Brehm et al.[25] discovered an unusual ferroelectric characteristic of CIPS, i.e., a uniaxial quadruple-potential well for Cu$^{1+}$ displacements. They theoretically and experimentally demonstrated that the low-polarization (LP) and high-polarization (HP) states correspond to the Cu$^{1+}$ displacement within and between vdW layers, respectively. In the metastable HP phase, the $c$-axis decreases from 13.4834 to 12.9305 Å, with an energy of 14 meV per Cu higher than the LP state. In the HP phase, the spontaneous polarization is greatly enhanced (~12.24 μC cm$^{-2}$) compared to the ground state (~3.34 μC cm$^{-2}$ with full intralayer sites occupancy of Cu$^{1+}$). On the other hand, giant negative piezoelectricity and nonlinear electrostriction in CIPS also suggest a strong correlation between spontaneous polarization and strain[26–28]. These earlier reports prompted us to investigate the spontaneous polarization of CIPS under pressure.

Figure 1b shows a schematic of the experimental setup. Single-crystal CIPS of ~10 μm thick is placed on a plastic holder and electrically connected to the ferroelectric tester through copper wires. Figure 1c illustrates the macroscopic polarization-electric field (P-E) hysteresis loops measured at different frequencies under ambient conditions, and the horizontal shift of the P-E loops with increasing frequency indicates defect dipoles in CIPS[29]. Figure 1d, e shows the P-E loops measured at 1 kHz under different pressures, and the change in remanent polarization (P$_r$) is summarized in Fig. 1f. A clear increase of P$_r$ up to 0.26 GPa is observed, which is then maintained at about 6.30 μC cm$^{-2}$ between 0.26 and 1.40 GPa. Subsequently, the remanent polarization gradually decreases upon further pressure increases, eventually disappearing at about 2.68 GPa. Supplementary Fig. S1 shows the P-E loops measured at 500, 200, 100, and 50 Hz, respectively, which reveal the same behavior.

### Understanding the enhancement of P$_r$ from lattice vibration

Previous studies[30,31] reported that CIPS maintains the ambient conditions phase up to 4–5 GPa then a phase transition from the monoclinic

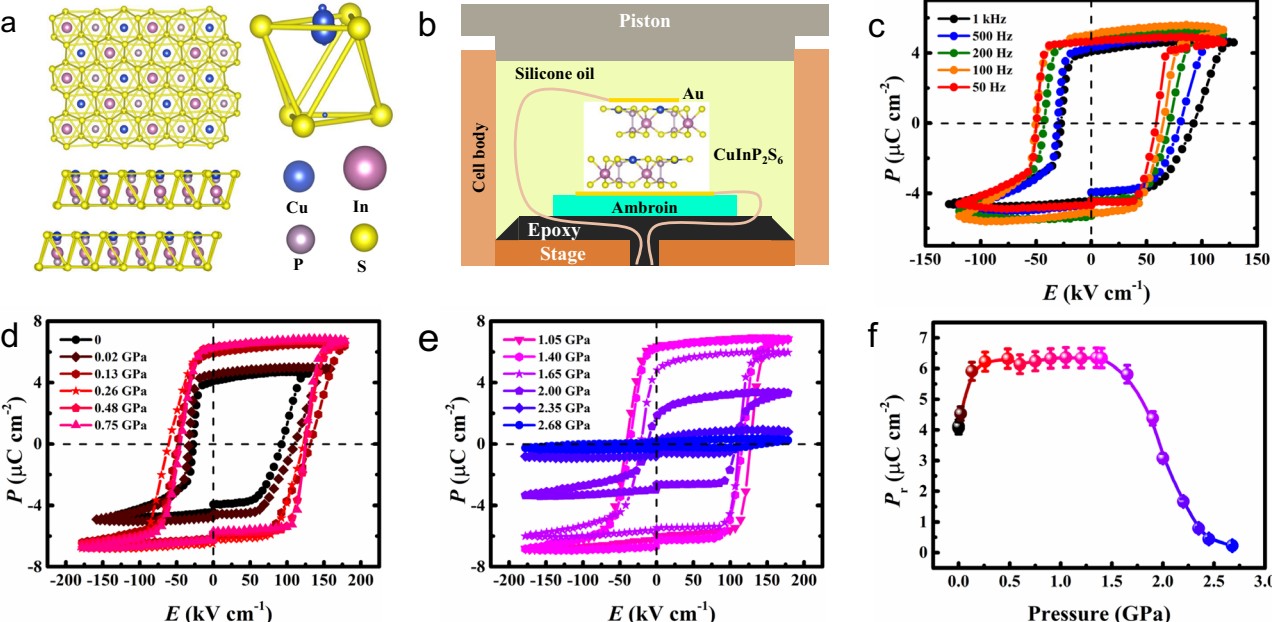

**Fig. 1 | P-E loops under pressure. a** Crystal structure of CIPS viewed along the layer normal direction ($c^*$ axis) and $b$ axis, respectively, with an octahedral sulfur cage showing various copper sites at room temperature. The size of the Cu atom represents its occupation probability. **b** Schematic of the high-pressure experimental setup. A CIPS ferroelectric crystal is held in a piston pressure cell. The yellow line embedded in epoxy represents an enameled copper wire. **c** Polarization-electric field (P-E) hysteresis loops of CIPS measured at different frequencies under ambient conditions. **d**, **e** P-E loops of CIPS measured at 1 kHz under representative pressures. **f** Remanent polarization (P$_r$) as a function of pressure, obtained from **d** and **e**. The error bars were estimated from the electrode areas.

($C_c$) to centrosymmetric trigonal ($P\bar{3}1m$) structure occurs. However, the sharp increase in spontaneous polarization up to about 0.26 GPa has not been reported. To elucidate the origin of this polarization enhancement, we explored the structural evolution under pressure using Raman spectroscopy. The Raman spectra of CIPS at different polarization states under ambient conditions were first studied. As shown in Supplementary Fig. S2, the Raman peaks of samples in polarization up and polarization down states are identical, as expected. The Raman spectrum of pristine CIPS before ferroelectric test is also shown in Supplementary Fig. S2, which serves as the reference for further analysis under pressure. There were no obvious changes in the topography of the sample after the pressure tests (Supplementary Fig. S3).

The high-pressure Raman spectroscopy results for CIPS are shown in Fig. 2a. The Raman peaks can be attributed to different vibration modes of CIPS[23,32], some of which are schematically shown in Fig. 2d. For example, the 72 cm$^{-1}$ peak arises from the out-of-plane Cu displacements and out-of-plane S vibration (black arrows); the 104 cm$^{-1}$ peak is attributed to rigid out-of-plane displacement of P-P dimers with In displacements opposite to that of the P-P dimers (red arrows); the 162 cm$^{-1}$ peaks is attributed to out-of-plane P-P and in-plane S vibration in ref. 32, but described as the change in S-P-P bond angle in ref. 23 ($\delta$(S-P-P)); similarly, the 216 and 238 cm$^{-1}$ peak are attributed to in-plane Cu + P, and out-of-plane S vibration in ref. 32, but described as the S-P-S bond angle change in ref. 23 ($\delta$(S-P-S)); the 264 cm$^{-1}$ peak arises from in-plane S vibration (light magenta arrows). The complete list is shown in Supplementary Table S1 in the Supplementary Information, which is adopted from ref. 32.

Figure 2b shows the logarithms of relative intensity ratios of representative Raman peaks as a function of pressure. Below 0.26 GPa, the intensities of the Cu$^{1+}$ vibrations peak at 72 cm$^{-1}$ gradually increases with pressure, while the FWHM value gradually decreases, indicating the location dispersion of Cu cations becomes narrower. The increase in the intensity of the 104 cm$^{-1}$ peak indicates that the distortion of $P_2S_6^{4-}$ cage decreases and becomes more uniform throughout the sample. As the $P_2S_6^{4-}$ cluster has a relatively rigid structural frame, its ordering drives Cu cations to the interlayer sites[23]. The 116 cm$^{-1}$ and 264 cm$^{-1}$ peaks show a slightly larger turning point in pressure as compared to other peaks. According to Supplementary Table S1, we can see that these two vibration modes are independent of the of P displacement, while majority of the other peaks involve P in some way. In CIPS, the S frame is mainly supported by P-P pair along the c-axis, which changes much more significantly under hydrostatic pressure than that of the a-b plane (see discussion below). We thus speculate that the peaks involve P would be more sensitive to small pressure increase than those that do not, such as the 116 and 264 cm$^{-1}$ peaks. For the 72 cm$^{-1}$ peak, even though it does not involve P displacement either, its main contribution comes from the out-of-plane movement of Cu cations, which is again very sensitive to pressure. Figure 2c shows the evolution of the relative Raman shift of representative modes. All modes are generally blue-shifted, except for modes at 216 and 238 cm$^{-1}$ under small pressure. The softening of these modes indicates that the bond angle deformation of S-P-S is weakened, which can be explained by the stiffness increase of the vibration through the enhanced interaction between Cu cations and adjacent S atoms, similar to what happens during the

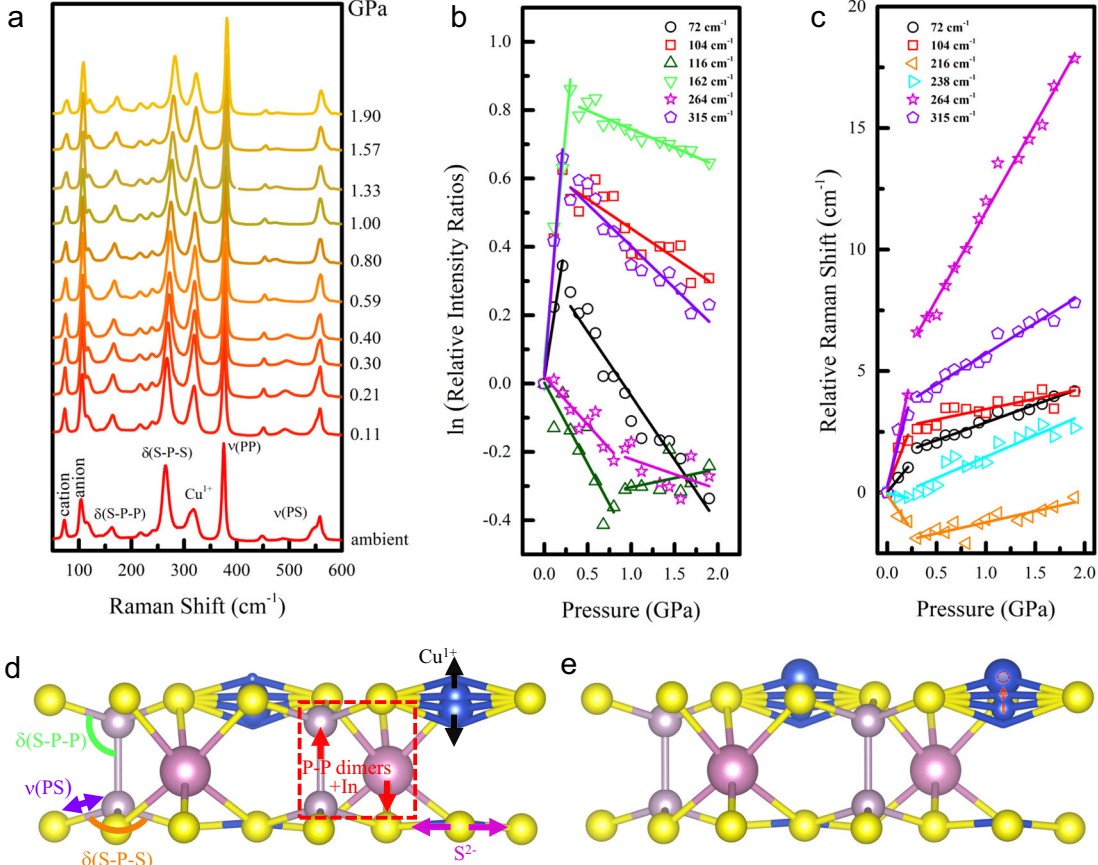

**Fig. 2 | Lattice vibration of CIPS. a** Pressure-dependent Raman spectra of CIPS from 0 to 1.90 GPa. **b** Pressure-evolution of the logarithms of the relative intensity ratio of the modes at 72, 104, 116, 162, 264, and 315 cm$^{-1}$. **c** Relative Raman shifts of the modes at 72, 104, 216, 238, 264, and 315 cm$^{-1}$ under pressure. **d** The schematic diagram of the Raman modes for cation, anion ($P_2S_6^{4-}$), S-P-P, S-P-S vibrations, P-P stretching, and P-S oscillations. **e** The schematic diagram of the evolution of Cu cations below 0.26 GPa.

paraelectric-ferroelectric transition in CIPS[23]. These Raman spectroscopy results suggest that, during the initial stage of increasing pressure, the Cu cations may be concentrating into the interlayer sites, resulting in the Raman peaks changes and the increased spontaneous polarization. This process is schematically illustrated in Fig. 2e.

## The competition between cell volume reduction and migration of Cu cations

Another possible contribution to the observed polarization enhancement is the cell volume reduction under pressure. To evaluate this effect quantitively, we performed in situ high-pressure single-crystal X-ray diffraction (SCXRD) measurements (Supplementary Figs. S4 and S5). Figure 3a shows the evolution of unit cell volume with pressure, a third-order Birch-Murnaghan equation of state was used to fit the data[33]. The fitting results give that $K_0$ (the isothermal bulk modulus) is 7.90(3) GPa, the $V_0$ (the initial volume) is 846.44(1) Å$^3$, and the $K_0'$ is 7.65(4). The results show that, if we assume that the dipoles do not change under pressure, the volume reduction would increase the macroscopic polarization by only about 5 % (0.12 μC cm$^{-2}$) at 0.26 GPa. The SCXRD results also reveal that c-axis compression mainly originates from the reduction of the vdW gaps, consistent with previous report[24]. Due to the enhanced coupling between Cu 4s and S sp orbitals in the adjacent layer as the interlayer distance reduces, it would be expected that the interlayer sites are more energetically favored[25] with the pressure increase, consistent with the Raman results shown in Fig. 2. Based on our SCXRD results and P-E loops obtained under pressure, we can also estimate the macroscopic negative piezoelectric coefficient ($e_{33} = \frac{\partial \mathbf{P}}{\partial \varepsilon}$, where $\mathbf{P}$ is the spontaneous polarization, and $\varepsilon$ is the c-axis strain) of CIPS upto

0.26 GPa, which is $\frac{2.30\,\mu\mathrm{C\,cm^{-2}}}{-0.00869} = -264\,\mu\mathrm{C\,cm^{-2}}$. This value is close to that reported in ref. 24 (−272 μC cm$^{-2}$).

Since the polarization enhancement is mainly due to increased occupancy of Cu cations at the interlayer sites, which is caused by the enhanced coupling between Cu 4s and S sp orbitals in the adjacent layer, we thus tried to estimate the percentage of Cu cations at different sites under pressure. We took the sites reported in ref. 24 as the possible locations of Cu cations. For Cu$^{1+}$ at different sites, the values of the corresponding dipoles were first calculated as described in the Methods. Under ambient conditions, detailed XRD analysis suggested (Fig. 3b inset) 32% Cu$^{1+}$ at sites Cu(1), 37.3% at sites Cu(2), 7.9% at sites Cu(3), 12% at sites Cu(4), there was also 8% at sites Cu(6)[24]. Starting from this distribution, we obtain a spontaneous polarization of 3.50 μC cm$^{-2}$. Considering that polarization back-switching occurred in part of the sample before XRD test, this value is consistent with our experimental result (P-E loops) under ambient conditions. As pressure increases, we suggest that the Cu cations migrate from the original sites (proportionally) into the interlayer sites Cu(4). As shown in Fig. 3b, the occupancy of interlayer sites reaches the maximum at 0.26 GPa and the change of Cu(4) sites occupancy is ~30%. At 2.00 GPa, the polarization is 2.47 μC cm$^{-2}$ with a zero occupancy of interlayer sites. The evolution of Cu(4) sites occupancy is summarized in Fig. 3b and schematically shown in Fig. 3c–f. To summarize, within the relatively low pressure range, the pressure mainly affects the interlayer distance of the material, reducing the relative energy of Cu interlayer sites and resulting in the enhancement of polarization. On the other hand, under higher pressure, the off-center displacement of Cu gradually decreases, similar to the behavior of Ti$^{4+}$ in PbTiO$_3$ and BaTiO$_3$ under pressure[17,18]. The short-range repulsions which prefer the

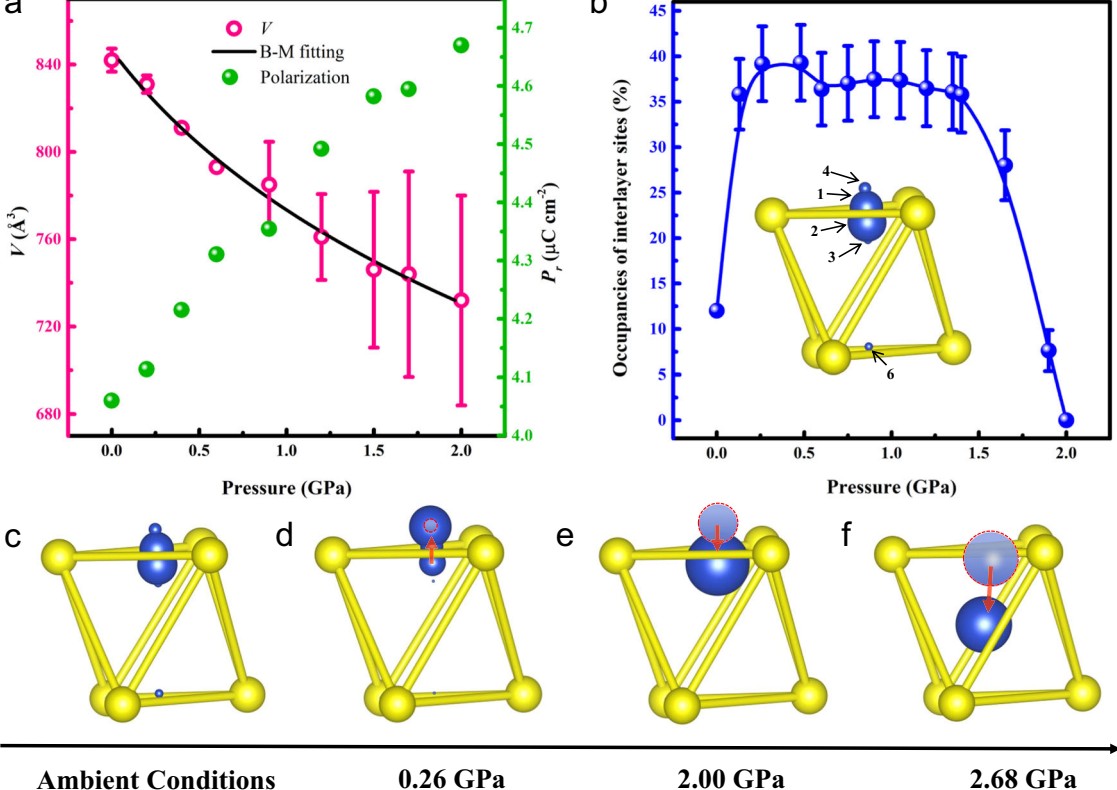

**Fig. 3 | Copper migration under pressure.** Pressure-dependent evolution of unit cell volume (**a**) and the occupancies of interlayer sites (**b**) from 0 to 2.00 GPa. The black line in **a** is the fitting of the pressure-cell volume curve and the green dots are the contribution of cell volume reduction to polarization increase, the error bars were estimated from the APEX3 software. **c** Highly smeared distribution of Cu atoms at ambient conditions. The size of Cu atoms represents the occupancy of different sites. **d** Below 0.26 GPa, the Cu atoms move toward interlayer sites according to the ferroelectric hysteresis loops measurements and Raman experimental results. **e** The atomic sites occupation of Cu is close to the ground state at 2.00 GPa. **f** The Cu atoms are completely located in the center of the S octahedron at 2.68 GPa, and ferroelectricity completely disappears.

undistorted paraelectric increase more rapidly than long-range attractions which favor ferroelectric distortions as pressure increases, leading to the suppression of ferroelectricity[34].

## Discussion

The Raman and SCXRD results suggest that polarization enhancement in CIPS below 0.26 GPa is related to the displacement of Cu cations with a small contribution from pure cell volume reduction effect. At high pressure (>1.40 GPa), ferroelectricity is gradually suppressed and completely disappears at 2.68 GPa, similar to traditional perovskite oxide ferroelectrics[17]. The off-center Cu cations displacements in CIPS result from the coupling of the chemically-active valence $d$ band with the $s$-like conduction band[35]. However, the increased optical band gap under high pressure indicates that the coupling is weakened, and the migration of Cu cations to the center of the S octahedron becomes the dominant factor[25]. The competition between the two factors likely give rise to the plateau of remanent polarization between 0.26 and 1.40 GPa.

To conclude, we have quantitatively determined the evolution of the remanent polarization of the vdW layered ferroelectric CIPS under pressure. An anomalous polarization enhancement was observed under low pressure, which stems from the spatial instability of Cu cations and the vdW layered structure. This result improves our understanding of the unique behaviors of vdW ferroelectric materials.

## Methods

### Sample preparation

High-quality single crystals of CIPS were grown by the chemical vapor transport method without a transport agent. Copper powder, indium powder, red phosphorus, and sulfur were placed in a quartz tube according to stoichiometric ratios. The temperature of the evaporation and crystallization zones was set to 750 °C and 650 °C, respectively, and the entire growth process lasted for 5 days.

### Ferroelectric measurements under pressure

To facilitate the testing of **P-E** loops, Au electrodes were deposited using direct current sputtering. The areas of electrodes range from 0.0095 to 0.0113 mm$^2$. The standard ferroelectric measurements were carried out with devices placed in a piston-cylinder pressure cell. Ferroelectric hysteresis loops were recorded using a commercial ferroelectric tester (Precision Multiferroic, Radiant Technologies).

### In situ Raman measurements under high pressure

A pair of diamond anvils with a flat top of 1000 μm was used, and a CuBe gasket was pre-indented to ~80 μm in thickness. A sample hole of ~500 μm was drilled in the center of the pre-indented gasket. Silicone oil was used as a pressure-transmitting medium (PTM) to ensure hydrostatic pressure conditions. The pressure was gauged at room temperature by monitoring the shift of the ruby R1 fluorescence line[36]. In situ high-pressure Raman experiments were carried out using a SpectraPro HRS-500 spectrometer with excitation lasers with a wavelength of 532 nm.

### In situ SCXRD under high pressure

For the SCXRD measurements, a pair of diamond anvils with a flat top of 500 μm was used, and a T-301 stainless-steel gasket was pre-indented to ~40 μm in thickness. The PTM also was silicone oil. In situ high-pressure SCXRD experiments were performed on a Bruker D8QUEST diffractometer with Mo Kα radiation (λ = 0.71073 Å). Diffraction data were collected by ω- and φ- scan methods, and lattice parameters and volumes were determined using APEX3 software.

### Surface characterizations

Atomic force microscopy (AFM) was performed on a Bruker Dimension Icon AFM equipped with a Nanoscope V controller. All measurements were performed under ambient conditions using the tapping mode.

### Numerical calculations

To calculate the polarization, we used the point charge model[3], where $\mathbf{P} = \frac{1}{V}(-e \sum N_i z_i)$, $V$ is the unit cell volume, $e$ is the electron charge, $N_i$ is the ionic valence state, $z_i$ is the projection of atomic position vector in a unit cell along the z direction. The reduced atom model is shown in Supplementary Fig. S6. The $Cu^{1+}$ shifts downward and the $In^{3+}$ and $P^{4+}$-$P^{4+}$ pair shift upward. The positive charge center is located at the atomic position ($z_i$) of Cu, In and P-P pairs, respectively, and the negative charge center is located at the average position of the six S ions. We calculated the electric dipoles by using the relative negative and positive charge positions, and divided S charges proportionally to Cu, In and P forming dipoles, respectively. The atomic positions are shown in Supplementary Table S2. The unit cell volume is $838.1931 \times 10^{-24}$ cm$^3$.

## Data availability

The data that support the findings of this study are available from the corresponding author upon request.

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

## Acknowledgements
The work was partly supported by the National Key R&D Program of China grant 2018YFA0305703, the National Natural Science Foundation of China grant No. 12274193, 12074164, 12004161 and 11904281, the Stable Support Plan Program of Shenzhen Natural Science Fund under grant No. 20200925152415003, the Guang dong Basic and Applied Basic Research Foundation of 2022A1515010044. J.Z. and Y.L. also acknowledged the Major Science and Technology Infrastructure Project of Material Genome Big-science Facilities Platform supported by Municipal Development and Reform Commission of Shenzhen. Some experiments were supported by the Synergic Extreme Condition User Facility. J.W. also acknowledges support from the Guangdong Provincial Key Laboratory Program (2021B1212040001) from the Department of Science and Technology of Guangdong Province, and the startup grant from the Southern University of Science and Technology (SUS-Tech), China.

## Author contributions
J.Z. and J.W. initiated this work. Y.B. prepared the samples of CIPS. X.Y. carried out the ferroelectric and Raman experiments under pressure with help from Q.Z. and Y.L., C.J. performed SCXRD. Z.X., L.C. and S.W. provided experimental equipment support. X.Z. participated in the discussion of experimental results. X.Y. wrote the first draft, X.Y., J.W. and J.Z. co-rewrote the manuscript.

## Competing interests
The authors declare no competing interests.
