## [Peer Review File · Nature Communications]

Anomalous polarization enhancement in a van der Waals ferroelectric material under pressureREVIEWER COMMENTS

Reviewer #1 (Remarks to the Author):

1. Please explain why silicone oil is used in the experiment. Is this the standard apparatus for this kind of experiment?
2. What does the merging of peaks at 549 cm⁻¹ and 556 cm⁻¹ mean to the structure change?
3. Fig. 2b and 2c need to be revised. Both figures share the same color between the two peaks, making it hard to distinguish the trend of the two separated peaks.
4. Please explain why peaks at 264 cm⁻¹ and 104/116 cm⁻¹ (I cannot tell due to the same color coding in Fig. 2b) have a different turning point in trend than the other peaks.
5. The authors should comment on the significantly increased volume deviation upon increasing pressure (especially when pressure approaches 1GPa) in fig.3a.
6. The line color in fig.1d and 1e is hard to distinguish, e.g., 0-0.13GPa, 0.26-0.75GPa seem share same color.
7. Please comment on why the intensity of peaks at 264 cm⁻¹ and 116 cm⁻¹ decrease upon increasing pressure up to ~0.75 GPa in fig. 2b.
8. The author indicates that the pressure-induced suppression of ferroelectricity is opposite to the usual behavior of other materials. However, the materials are different, and this method only seems effective to CIPS. Does this imply that the finding is exclusively for CIPS?
9. But in 1e, the material has a smaller ferroelectricity in growing pressure. And eventually goes to almost zero ferroelectricity. This is saying there are ranges of pressure that could enhance and decrease the ferroelectricity. The pressure range of enhancing the ferroelectricity is quite small. This small enhancement does not indicate a significant improvement in performance.
10. The Cu ion's migration is confusing to the reviewer. Why is the 0.26 GPa so special? At this point, the Cu ion is moving up below this point, while beyond this point, the Cu ion is moving downwards.

Reviewer #2 (Remarks to the Author):

The stress-induced phase transitions and strain engineering become especially important in nanoparticles and ultrathin films of Cu-based layered chalcogenides, CuInP₂(S,Se)₆; these materials are uniaxial ferroics, whose value for advanced applications is due to a possibility of the ferroelectricity and antiferroelectricity downscaling to the limit of a single layer. Despite the significant fundamental and practical interest in bulk and nano-sized CuInP₂(S,Se)₆, the influence of stress and strains on its spontaneous polarization is very poorly studied from experimental perspective, while some theoretical background is available. Hence, the experimental part of the peer-reviewed manuscript, reporting about anomalous polarization enhancement in a vdW ferroelectric material under pressure is original, well-written and perfectly illustrated. I regard that it deserves publication in Nature Communications. However, I have a principal remark regarding possible theoretical interpretation of the revealed polarization enhancement. Authors should semi-quantitatively compare the experimentally observed polarization enhancement with earlier theoretical predictions (see figures 3 and 4 in [A. N. Morozovska, et al. Stress-Induced Phase Transitions in Nanoscale CuInP₂S₆. Phys. Rev. B 104, 054102 (2021) <https://link.aps.org/doi/10.1103/PhysRevB.104.054102>]). From a theoretical viewpoint, the polarization enhancement occurs due to the strong, negative, and temperature-dependent nonlinear electrostriction coupling coefficients (namely $Z_{i33}<0$ and $W_{ij3}<0$), and the "inverted" signs of the linear electrostriction coupling coefficients (namely $Q_{33}<0$, $Q_{23}>0$, and $Q_{13}>0$) for CIPS. The hydrostatic pressure effect on the polarization enhancement is

complex and unusual in comparison with many ferroelectrics where $Q_{33} > 0$, $Q_{23} < 0$, and $Q_{13} < 0$.

Reviewer #3 (Remarks to the Author):

Yao et al. present novel results in the effect of hydrostatic pressure on the ferroelectricity of CuInP_2S_6 (CIPS) and its crystallographic structure arrangement under such conditions. The experimental results are of good quality and the conclusions of the study might help engineer electronic and ionic properties of CIPS and other vdW ferroelectrics for device applications. I would recommend the article for publication after minor changes/comments are added or explained:

1. How does the hydrostatic pressure affect the domain distribution in the material? PFM in liquid is not trivial to perform (if even possible) but maybe micro-Raman spectroscopy could give a hint. Even in-situ AFM topography images of the surface of the sample should help gathering information about that. This information would be interesting to answer the following 2 question:

1.1 Does the hydrostatic pressure switch the material into one of the polarization states (either up or down)? If so, maybe in the intermediate range of pressures (0.24-1.5GPa) where the enhancement of the polarization is found, polarization is being switched or Cu ion position is being moved from LP to HP polarization states. The nanoscale distribution of the domains in normal conditions and under pressure should give a hint.

1.2 Is the surface affected/damaged by the hydrostatic pressure? (like it happens in 1 and others).

Some surface characterization should be added or at least commented in the text for clarification.

2. Continuing with the importance of the nanoscale domain distribution for the overall measured response of the material, the pressure ranges studied in this work are accessible with AFM. Therefore, the polarization enhancement could be addressed by performing PFM measurements at different force setpoints, similarly to how the authors performed in 2. In that case, the authors attributed the switching to flexoelectric interaction due to the non-uniform pressure exerted by the AFM tip, generating a strain gradient instead of a homogeneous strain.

How can we separate both distributions? Performing PFM with varying tip radius and setpoints should directly affect the generated strain gradient fields, therefore accessing the deconvoluted information between strain and strain gradient being the responsible of the enhanced/suppressed ferroelectric response.

Such characterization should be added or at least commented in the text for clarification.

3. When synthesized with Cu deficiencies, a Cu free non-ferroelectric phase (IPS) forms in the material. Cu movement has been found to be enhanced at the CIPS/IPS phase boundaries³. In addition, other studies also show that Cu migration is highly anisotropic in such materials⁴. Does the hydrostatic pressure induce any sort of Cu migration in the lateral direction for pure CIPS? Does the hydrostatic pressure enhance/induce Cu migration laterally in CIPS/IPS phase boundaries?

4. How is the Curie temperature of the material affected by the hydrostatic pressure? Measurements of the phase transition at different hydrostatic pressures should allow obtaining a more complete phase diagram of the material, confirming the Cu ion position shown in this work.

References:

- 1 Rao, R., Conner, B. S., Selhorst, R. & Susner, M. A. Pressure-driven phase transformations and phase segregation in ferroelectric CuInP_2S_6 - $\text{In}_4/3\text{P}_2\text{S}_6$ self-assembled heterostructures. *Physical Review B* 104, 235421 (2021).
- 2 Ming, W. et al. Flexoelectric engineering of van der Waals ferroelectric CuInP_2S_6 . *Science Advances* 8 (2022). <https://doi.org/doi:10.1126/sciadv.abq1232>
- 3 Checa, M. et al. Revealing Fast Cu-Ion Transport and Enhanced Conductivity at the CuInP_2S_6 - $\text{In}_4/3\text{P}_2\text{S}_6$ Heterointerface. *ACS nano* 16, 15347-15357 (2022).
- 4 Zhang, D. et al. Anisotropic Ion Migration and Electronic Conduction in van der Waals Ferroelectric CuInP_2S_6 . *Nano letters* 21, 995-1002 (2021).

Responses to Reviewers' Comments

(Research Article, NCOMMS-23-08192A)

We thank the reviewers for their valuable comments. The following are our point-by-point responses.

Responses to Reviewer #1's comments

1. Please explain why silicone oil is used in the experiment. Is this the standard apparatus for this kind of experiment?

Authors' reply: Thanks for this comment. In high-pressure experiments, pressure-transmitting mediums (PTM) are used to achieve hydrostatic pressure condition. The common PTMs include silicone oil, glycerol, 4:1 methanol-ethanol, and Daphne 7377. In this work, silicone oil does not react with our sample, and was chosen as the PTM.

2. What does the merging of peaks at 549 cm^{-1} and 556 cm^{-1} mean to the structure change?

Authors' reply: We appreciated the reviewer's comment. Looking closely at the spectra, one would notice that the two peaks do not simply merge together, but rather the 549 cm^{-1} peak shifts gradually to the right of the 556 cm^{-1} peak. The trend is similar to some of the other peaks, but because of its relatively low intensity and close proximity to the 556 cm^{-1} peak, we didn't choose this peak for further analysis. Note that both the 549 and 556 cm^{-1} peaks are attributed to P-S oscillations ^[1].

3. Fig. 2b and 2c need to be revised. Both figures share the same color between the two peaks, making it hard to distinguish the trend of the two separated peaks.

Authors' reply: Thank you for your suggestion. We have updated it in the revised manuscript.

4. Please explain why peaks at 264 cm^{-1} and $104/116\text{ cm}^{-1}$ (I cannot tell due to the same color coding in Fig. 2b) have a different turning point in trend than the other peaks.

Authors' reply: Thanks for this comment. We have changed the colors in the revised manuscript to make them distinguishable. The 116 cm^{-1} peak is attributed to the in-plane displacement of Cu + In + S and the out-of-plane S vibration and the 264 cm^{-1} peak is attributed to the in-plane S vibration. ^[2] These peaks are independent of the displacement of P atoms, while majority of the other peaks involve P in some way. In CIPS, the S frame is mainly supported by P-P dimers along the *c*-axis, which changes much more significantly under hydrostatic pressure than that of the *a-b* plane. We thus speculate that the peaks involve P would be more sensitive to pressure increase than those that do not, such as the 116 and 264 cm^{-1} peaks. However, further theoretical analysis would be needed to confirm this hypothesis.

To better understand the change in Raman spectra and structural evolution under pressure, we have established a one-to-one correlation between the Raman peaks and vibration modes in CIPS as shown in Table R1 following reference [2], taking into consideration the conclusions of reference [1].

Raman shift	Displacement patterns
72 cm ⁻¹	out-of-plane Cu (polar for A' and antipolar for A'' displacements in the adjacent layers) + out-of-plane S vibration
104 cm ⁻¹	rigid out-of-plane displacement of P-P dimers (in-phase phase in adjacent layers) + In displacements opposite to that of P-P dimers
116 cm ⁻¹	in-plane displacement of Cu + In + S, out-of-plane S vibration
162 cm ⁻¹	out-of-plane P-P + in-plane S vibration
216 cm ⁻¹	in-plane Cu + P, and out-of-plane S vibration
238 cm ⁻¹	in-plane Cu + P, and out-of-plane S vibration
264 cm ⁻¹	in-plane S vibration
315 cm ⁻¹	in-plane P + S vibration
375 cm ⁻¹	out-of-plane P + in-plane S vibration
448 cm ⁻¹	out-of-plane P + out-of-plane S vibration
549 cm ⁻¹	in-plane P-P stretching + in-plane S vibration
558 cm ⁻¹	in-plane P-P + in-plane S vibration

Table R1 Raman peaks and vibration modes in CIPS [2]

Page 6 of the main text is revised and Figure 2 redrawn accordingly, but the main conclusions remain the same. Table R1 is also added to the Supplementary Information for reference.

5. The authors should comment on the significantly increased volume deviation upon increasing pressure (especially when pressure approaches 1GPa) in fig.3a.

Authors' reply: Thanks for this comment. The number of diffraction spots that can be collected decreases with pressure, as shown in Figure S7 in the supplementary material. With less diffraction spots for the refinement, there is the significantly increased error bar upon increasing pressure.

6. The line color in fig.1d and 1e is hard to distinguish, e.g., 0-0.13GPa, 0.26-0.75GPa seem share same color.

Authors' reply: Thank you for your suggestion. We have updated it in the revised manuscript.

7. Please comment on why the intensity of peaks at 264 cm^{-1} and 116 cm^{-1} decrease upon increasing pressure up to $\sim 0.75\text{ GPa}$ in fig. 2b.

Authors' reply: Thanks for this comment. The decrease of 264 cm^{-1} peak intensity has been reported before. For example, Vysochanskii et al have observed that the relative intensity of 264 cm^{-1} peak gradually decreases when temperature drops, as shown in Ref [1] figure 3. ^[1] Both peaks involve in-plane movement of S, which is strongly affected by the Cu ions. We suggest that, with decreasing temperature or increasing pressure, the dispersion of Cu cations gradually decreases, leading to a more stable S frame which is more resistant to in-plane distortion, thus the intensities both peaks drop.

8. The author indicates that the pressure-induced suppression of ferroelectricity is opposite to the usual behavior of other materials. However, the materials are different, and this method only seems effective to CIPS. Does this imply that the finding is exclusively for CIPS?

Authors' reply: Thanks for this comment. Indeed, the unusual enhancement of ferroelectricity in CIPS is directly related to its unique vdW structure. The strong effect of pressure on the c -axis lattice alters the interaction of cations (Cu) within one layer with the anions (S) in the next layer, changing the location of cations and the macroscopic polarization. However, with more and more vdW ferroelectric materials being discovered, we do expect that similar behavior may be observed in other vdW materials, e.g. sliding ferroelectrics. ^[3]

Furthermore, it has been reported that the ferroelectricity of multiferroics CuCrO_2 and TbMnO_3 ^[4,5] enhances within a certain pressure range, in which case the mechanism was attributed to the pressure-induced magnetoelectric phase transition.

To describe the background more accurately, we added the following sentences on page 3 in the revised manuscript. "For example, the spontaneous polarizations of PbTiO_3 and BaTiO_3 at room temperature were totally suppressed under pressures of 10 GPa and 2 GPa, respectively. ^{17,18} Though it has been reported that the ferroelectricity of multiferroic CuCrO_2 and TbMnO_3 ^{19,20} enhances within a certain pressure range, but they were attributed to pressure-induced magnetoelectric phase transitions. The enhancement of remanent polarization in vdW ferroelectric CIPS is another example worthy of further investigation."

9. But in 1e, the material has a smaller ferroelectricity in growing pressure. And eventually goes to almost zero ferroelectricity. This is saying there are ranges of pressure that could enhance and decrease the ferroelectricity. The pressure range of enhancing the ferroelectricity is quite small. This small enhancement does not indicate a significant improvement in performance.

Authors' reply: We appreciated the reviewer's comment. It is the unusual behavior that is interesting here. In conventional ferroelectric materials with continuous three-dimensional lattice, i.e., except type-II multiferroics where the polarization arises from certain magnetic order, hydrostatic pressure will suppress lattice distortion and the ferroelectricity. This remains to be true for CIPS under high pressure. However, the existence of vdW gaps changes the low pressure behavior completely. We believe that clarifying the mechanism behind offers valuable information for bettering understanding of the growing family of vdW ferroelectrics.

10. The Cu ion's migration is confusing to the reviewer. Why is the 0.26 GPa so special? At this point, the Cu ion is moving up below this point, while beyond this point, the Cu ion is moving downwards.

Authors' reply: Thanks for this comment. The 0.26 GPa is not so special. The pressure changes the lattice and the relative energy of Cu at different sites, resulting in the redistribution of Cu. For CIPS, due to its particular mechanical properties such as elastic modulus, it happens to reach the maximum polarization (metastable state) at 0.26 GPa. Perhaps with a different material such as CuInP_2S_6 , the maximum polarization may not be at 0.26 GPa.

At high pressure, the pressure dependence of polarization is similar to BaTiO_3 and PbTiO_3 .^[6,7] The widely used theory suggests that short-range repulsions increase more rapidly than long-range attractions as pressure increases, leading to the reduction of ferroelectricity.^[8] The distance of Cu from the center of the S octahedron has been decreasing under pressure, so the Cu ion is moving downwards.

Accordingly, we added the following discussion on page 9 in the revised manuscript. "To summarize, within the relatively low pressure range, the pressure mainly affects the interlayer distance of the material, reducing the relative energy of Cu interlayer sites and resulting in the enhancement of polarization. On the other hand, under higher pressures, the off-center displacement of Cu gradually decreases, similar to the behavior of Ti^{4+} in PbTiO_3 and BaTiO_3 under pressure.^{17,18} The short-range repulsions which prefer the undistorted paraelectric increase more rapidly than long-range attractions which favor ferroelectric distortions as pressure increases, leading to the suppression of ferroelectricity.³⁴"

Reference:

- [1] Vysochanskii Y M, Stephanovich V A, Molnar A A, et al. Raman spectroscopy study of the ferroelectric-paraelectric transition in layered CuInP_2S_6 [J]. Physical Review B, 1998, 58(14): 9119.
- [2] Neal S N, Singh S, Fang X, et al. Vibrational properties of CuInP_2S_6 across the ferroelectric transition[J]. Physical

Review B, 2022, 105(7): 075151.

[3] Ding N, Chen J, Gui C, et al. Phase competition and negative piezoelectricity in interlayer-sliding ferroelectric ZrI_2 [J]. *Physical Review Materials*, 2021, 5(8): 084405.

[4] Aoyama T, Yamauchi K, Iyama A, et al. Giant spin-driven ferroelectric polarization in $TbMnO_3$ under high pressure[J]. *Nature communications*, 2014, 5(1): 4927.

[5] Aoyama T, Miyake A, Kagayama T, et al. Pressure effects on the magnetoelectric properties of a multiferroic triangular-lattice antiferromagnet $CuCrO_2$ [J]. *Physical Review B*, 2013, 87(9): 094401.

[6] Sani A, Hanfland M, Levy D. Pressure and temperature dependence of the ferroelectric–paraelectric phase transition in $PbTiO_3$ [J]. *Journal of Solid State Chemistry*, 2002, 167(2): 446-452.

[7] Venkateswaran U D, Naik V M, Naik R. High-pressure Raman studies of polycrystalline $BaTiO_3$ [J]. *Physical Review B*, 1998, 58(21): 14256.

[8] Samara G A, Sakudo T, Yoshimitsu K. Important generalization concerning the role of competing forces in displacive phase transitions[J]. *Physical Review Letters*, 1975, 35(26): 1767.

Responses to Reviewer #2

The stress-induced phase transitions and strain engineering become especially important in nanoparticles and ultrathin films of Cu-based layered chalcogenides, $\text{CuInP}_2(\text{S},\text{Se})_6$; these materials are uniaxial ferroics, whose value for advanced applications is due to a possibility of the ferroelectricity and antiferroelectricity downscaling to the limit of a single layer. Despite the significant fundamental and practical interest in bulk and nano-sized $\text{CuInP}_2(\text{S},\text{Se})_6$, the influence of stress and strains on its spontaneous polarization is very poorly studied from experimental perspective, while some theoretical background is available. Hence, the experimental part of the peer-reviewed manuscript, reporting about anomalous polarization enhancement in a vdW ferroelectric material under pressure is original, well-written and perfectly illustrated. I regard that it deserves publication in Nature Communications.

However, I have a principal remark regarding possible theoretical interpretation of the revealed polarization enhancement. Authors should semi-quantitatively compare the experimentally observed polarization enhancement with earlier theoretical predictions (see figures 3 and 4 in [A. N. Morozovska, et al. Stress-Induced Phase Transitions in Nanoscale CuInP_2S_6 . Phys. Rev. B 104, 054102 (2021) <https://link.aps.org/doi/10.1103/PhysRevB.104.054102>]). From a theoretical viewpoint, the polarization enhancement occurs due to the strong, negative, and temperature-dependent nonlinear electrostriction coupling coefficients (namely $Z_{i33}<0$ and $W_{ij3}<0$), and the “inverted” signs of the linear electrostriction coupling coefficients (namely $Q_{33}<0$, $Q_{23}>0$, and $Q_{13}>0$) for CIPS. The hydrostatic pressure effect on the polarization enhancement is complex and unusual in comparison with many ferroelectrics where $Q_{33}>0$, $Q_{23}<0$, and $Q_{13}<0$.

Authors' reply: We appreciated the reviewer's comments. According to Ref [1], the polarization of bulk CIPS increases by about 32.2 % from $3.1 \mu\text{C cm}^{-2}$ at ambient conditions to $4.1 \mu\text{C cm}^{-2}$ at 0.26 GPa, which is smaller than but comparable to ours observation of 56.5% enhancement.

Regarding the correlation between polarization enhancement with electrostriction, we argue that both are consequences/manifestations of microscopic structural changes, which is why we centered our analysis around the structural evolution as revealed by Raman spectroscopy.

Accordingly, the following statement was added on page 4 in the revised manuscript. “On the other hand, giant negative piezoelectricity and nonlinear electrostriction in CIPS also suggest a strong correlation between spontaneous polarization and strain.^{26,27,28}”

Reference:

[1] Morozovska A N, Eliseev E A, Kalinin S V, et al. Stress-induced phase transitions in nanoscale CuInP_2S_6 [J]. Physical Review B, 2021, 104(5): 054102.

Responses to Reviewer #3

Yao et al. present novel results in the effect of hydrostatic pressure on the ferroelectricity of CuInP_2S_6 (CIPS) and its crystallographic structure arrangement under such conditions. The experimental results are of good quality and the conclusions of the study might help engineer electronic and ionic properties of CIPS and other vdW ferroelectrics for device applications. I would recommend the article for publication after minor changes/comments are added or explained:

1. How does the hydrostatic pressure affect the domain distribution in the material? PFM in liquid is not trivial to perform (if even possible) but maybe micro-Raman spectroscopy could give a hint. Even in-situ AFM topography images of the surface of the sample should help gathering information about that. This information would be interesting to answer the following 2 question:

Authors' reply: Thanks for this comment. Unfortunately, it is not possible for us to look into the domain structure under pressure using PFM. We performed Raman study on samples that are polarized upward and downward, respectively, and observed no changes in the spectra.

Figure R1 Raman spectra of CIPS in the pristine (black line), downward-polarization (red line) and upward-polarization states (blue line).

Accordingly, the following statement was added on page 5 in the revised manuscript. “The Raman spectra of CIPS at different polarization states under ambient conditions were first studied. As shown in Figure S2, the Raman peaks of samples in polarization up and polarization down states are identical, as expected. The Raman spectrum of pristine CIPS before ferroelectric test is also shown in Figure S2, which serves as the

reference for further analysis under pressure (Raman spectra were collected from regions without top electrodes while ferroelectric loops were obtained on the same sample with top electrodes).”

The above figure is also added to the supplementary information as “**Figure S2**”.

1.1 Does the hydrostatic pressure switch the material into one of the polarization states (either up or down)? If so, maybe in the intermediate range of pressures (0.24-1.5GPa) where the enhancement of the polarization is found, polarization is being switched or Cu ion position is being moved from LP to HP polarization states. The nanoscale distribution of the domains in normal conditions and under pressure should give a hint.

Authors’ reply: Thanks for this comment. We do not expect that the hydrostatic pressure will switch the polarization. In principle, the uniform hydrostatic pressure does not favor one polarization state over the other. However, even if it does, it will not affect our experimental results. Note that the polarization values of CIPS under pressure were obtained by measuring the *P-E* hysteresis loops, and polarization will make a full 180 degrees switch during this process. The larger switchable polarization suggest that Cu ions are being moved to HP states.

1.2 Is the surface affected/damaged by the hydrostatic pressure? (like it happens in I and others). Some surface characterization should be added or at least commented in the text for clarification.

Authors’ reply: We appreciated the reviewer’s comment. We have carried out atomic force microscopy (AFM) characterizations on CIPS before and after the pressure test. We did not observe significant changes, as shown in Figure R2.

Figure R2 Surface analysis before and after the pressure test. (a) AFM topography of a pristine CIPS single crystal. The Ra that is arithmetic average of absolute values is 0.137 nm. (b) AFM topography of the decompressed sample, the Ra is 0.113 nm.

Accordingly, we added the sentence “There were no obvious changes in the topography of the sample of the pressure tests” on page 5 in the revised manuscript.

The above figure is added to the Supplementary Information as “Figure S3”.

2. Continuing with the importance of the nanoscale domain distribution for the overall measured response of the material, the pressure ranges studied in this work are accessible with AFM. Therefore, the polarization enhancement could be addressed by performing PFM measurements at different force setpoints, similarly to how the authors performed in 2. In that case, the authors attributed the switching to flexoelectric interaction due to the non-uniform pressure exerted by the AFM tip, generating a strain gradient instead of a homogeneous strain.

How can we separate both distributions? Performing PFM with varying tip radius and setpoints should directly affect the generated strain gradient fields, therefore accessing the deconvoluted information between strain and strain gradient being the responsible of the enhanced/suppressed ferroelectric response.

Such characterization should be added or at least commented in the text for clarification.

Authors’ reply: Thanks for this comment. In our experimental conditions, the material is in a uniform hydrostatic environment, so we expect the force to be isotropic and there is little or no strain gradient.

To distinguish these two experiments, we added the following sentences on page 3 in the revised manuscript. “Recently, Ming *et al.*²¹ reported an alternative switching method for CIPS using flexoelectric effect. In their experiments, strain gradient drives the migration of Cu cations. However, in this study, hydrostatic pressure is applied, promoting more Cu cations to occupy interlayer sites, resulting in the overall enhancement of polarization.”

3. When synthesized with Cu deficiencies, a Cu free non-ferroelectric phase (IPS) forms in the material. Cu movement has been found to be enhanced at the CIPS/IPS phase boundaries³. In addition, other studies also show that Cu migration is highly anisotropic in such materials⁴. Does the hydrostatic pressure induce any sort of Cu migration in the lateral direction for pure CIPS? Does the hydrostatic pressure enhance/induce Cu migration laterally in CIPS/IPS phase boundaries?

Authors’ reply: We appreciated the reviewer’s comment. Unfortunately, we are not able to study the lateral migration of Cu under pressure. However, because of the isotropic nature of the hydrostatic pressure, we do not expect Cu ions to migrate significantly in-plane. Our samples are phase pure CIPS as revealed in Figure R3 by the XRD and TEM results.

Figure R3 (a) The single-crystal X-ray diffraction (SCXRD) pattern of CIPS. (b) The selected area electron diffraction pattern by transmission electron microscopy (TEM-SAD) of CIPS, the pattern was recorded along the $\langle 001 \rangle$ zone axis. These results show that the samples are phase pure.

4. How is the Curie temperature of the material affected by the hydrostatic pressure? Measurements of the phase transition at different hydrostatic pressures should allow obtaining a more complete phase diagram of the material, confirming the Cu ion position shown in this work.

Reply: Thanks for this comment. As shown in Ref [1], the phase transition temperature increases with pressure at a rate of $210\text{K}/\text{GPa}$. According to their results, in the pressure range of $p > 0.4\text{ GPa}$, the anomaly of the dielectric permeability will be completely masked by copper ion conductivity.³ Due to the difficulty of high-temperature and high-pressure experiments, these experiments may be the focus of our next exploration.

References:

[1] Shusta V S, Prits I P, Guranich P P, et al. Dielectric properties of CuInP_2S_6 crystals under high pressure[J]. 2007.

REVIEWERS' COMMENTS

Reviewer #1 (Remarks to the Author):

The authors have addressed my comments.

Reviewer #2 (Remarks to the Author):

The Authors satisfactory replied on my remark, and I recommend the revised manuscript for publication in Nature Communications

Responses to Reviewers' Comments

(Research Article, NCOMMS-23-08192B)

We are delighted that all the reviewers are satisfied with our revision and reply to their comments, and no further comments of the manuscript entitled “Anomalous polarization enhancement in a vdW ferroelectric material under pressure” by Yao et al.

Responses to Reviewer #1's comments

1. The authors have addressed my comments.

Answer: Thank you.

Responses to Reviewer #2's comments

1. The Authors satisfactorily replied on my remark, and I recommend the revised manuscript for publication in Nature Communications.

Answer: Thank you.